# Warnings based on risk matrices: a coherent framework with consistent evaluation

Robert J. Taggart[1] and David J. Wilke[2]

[1]Bureau of Meteorology, Sydney, Australia
[2]Bureau of Meteorology, Melbourne, Australia

**Correspondence:** Robert J. Taggart (robert.taggart@bom.gov.au)

**Abstract.** Risk matrices are widely used across a range of fields and have found increasing utility in warning decision practices globally. However, their application in this context presents challenges, which range from potentially perverse warning outcomes to a lack of objective verification (i.e., evaluation) methods. This paper introduces a coherent framework for generating multi-level warnings from risk matrices to address these challenges. The proposed framework is general, is based on probabilistic forecasts of hazard severity or impact and is compatible with the Common Alerting Protocol (CAP). Moreover, it includes a family of consistent scoring functions for objectively evaluating the predictive performance of risk matrix assessments and the warnings they produce. These scoring functions enable the ranking of forecasters or warning systems and the tracking of system improvements by rewarding accurate probabilistic forecasts and compliance with warning service directives. A synthetic experiment demonstrates the efficacy of these scoring functions, while the framework is illustrated through warnings for heavy rainfall based on operational ensemble prediction system forecasts for Tropical Cyclone Jasper (Queensland, Australia, 2023). This work establishes a robust foundation for enhancing the reliability and verifiability of risk-based warning systems.

## 1 Introduction

Early warning systems are used globally to alert a range of users, including the general public, to act to reduce the potential for harm in advance of hazardous events (UNDRR, 2024). A key feature of many warning systems is the use of warning levels, which typically operate as a series of escalating categories used to communicate threat. Levels can be based on a range of information. Many warning levels are informed directly by the severity or urgency of a hazard (Neußner, 2021). For example, warnings from the German Weather Service (DWD) have four levels which correspond to increasing hazard strength (DWD, 2024). More recently, there has been a move by meteorological agencies toward impact-based forecasting (WMO, 2015), and many services now use levels to explicitly indicate escalating impact potential. The UK Met Office (UKMO) issues impact-based warnings with the highest level, 'Red Warning', corresponding to a very likely risk to life and substantial disruption (UK Met Office, 2024). This is an approach reflected by many emergency services agencies. For example, the warning system used in Australia matches the action required by the community to promote safety (AIDR, NEMA, 2024).

Determining when and at what level to issue a warning is a key decision ideally informed by a probabilistic assessment of possible outcomes made by a forecaster (or forecast system) in conjunction with well-defined criteria. Such criteria are

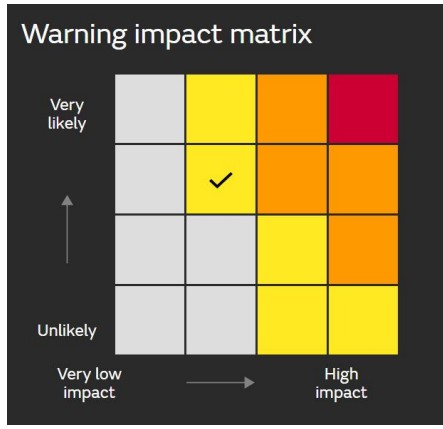

**Figure 1.** The warning impact matrix used by UKMO (UK Met Office, 2024). In this example a 'Yellow Warning' level is issued, as indicated by the 'tick'.

known as 'forecast directives' (Murphy and Daan, 1985) and should be determined in consultation with key stakeholders of the warning service. An example of a forecast directive for a warning service for damaging wind gusts is "Issue a warning if and only if the probability of a wind gust exceeding 90 km/h is at least 10%". Historically, considerable effort has been invested to advise or recommend thresholds that define severity categories (Stepek et al., 2012). In recent years, probability or likelihood information has been increasingly incorporated into warning decision frameworks. This shift recognises the need for an appropriate degree of confidence before issuing a warning, and is driven by a surge in probabilistic data following the advent of ensemble-based forecasting systems (Legg and Mylne, 2004). Over a decade ago, the UKMO pioneered the use of impact–likelihood matrices, an example of which is shown in Fig. 1, to communicate risk and generate warning levels for weather related hazards in the United Kingdom (Neal et al., 2014; Hemingway and Robbins, 2020; UK Met Office, 2024). This approach has since been endorsed by the World Meteorological Organisation (WMO, 2015) and is used by a number of other meteorological agencies (Indonesian Agency for Meteorology, Climatology, and Geophysics (BMKG), 2016; South African Weather Service, 2020; Barbados Meteorological Services, 2020). These 'risk matrices', which are widely used in a range of fields, have found increasing utility in warning decisions as they provide a simple process for determining the warning level based on the severity (or impact) and associated certainty (or likelihood) of the hazard. Ideally, the warning level increases as the hazard becomes sufficiently more severe and/or certain. When well-defined and coherent, such risk matrices can constitute a forecast directive for the warning service.

Despite their prevalence, serious problems have been identified with the conventional use of risk matrices in other fields (Cox, 2008; Ball and Watt, 2013). In the warning context, current operational risk matrix approaches also have problems, including situations where warning level decreases even as hazard likelihood increases, as discussed in Section 2.2. Furthermore, issues arise when categories are not well-defined, as is often the case with services based on impact or consequence which are usually described qualitatively. When warning decision process lacks adequate definition, two forecasters with identical probabilistic assessments of the hazard could issue two different warning levels. This may lead to warning messages that fluctuate

unnecessarily, compromising both public safety and service credibility. Furthermore, it also limits verification (i.e., evaluation) of warning performance post-event. In fact, while UKMO's impact-based warning system undergoes some subjective verification, objective verification is not conducted (Suri and Davies, 2021) and a recent attempt at more objective verification has encountered difficulties (Sharpe, 2024).

Warning verification is important for a number of reasons. Firstly, it enables agencies to monitor performance, compare the quality of different forecast systems, and implement targeted, evidence-based enhancements to improve service prediction (Loveday et al., 2024). Secondly, through objective verification, agencies can demonstrate their performance to users, thereby building trust. An indispensable tool for measuring the performance of warning accuracy, or for assessing whether a system upgrade will lead to improved prediction for a warning service, is a *scoring function* that is *consistent* for the prediction task at hand (Gneiting, 2011). A scoring function assigns a number to a forecast, which is a measure of how good the forecast was compared to the corresponding observation. In the context of warnings, a scoring function is consistent if the forecaster's expected score is optimised by following the forecast directive. The FIRM framework (Taggart et al., 2022) provides a family of scoring functions that are consistent for warning services that: (1) have warning levels that correspond to severity categories and (2) use a measure of certainty based on either a fixed probability threshold, or on the severity category in which the expected value lies. The FIRM score has recently been used to recommend substantial predictive improvements to a heatwave warning service (Loveday and Carroll, 2025). However, for warning services in which the warning level is determined using a risk matrix, consistent scoring functions have not been available, until now.

In this paper we present a general framework for determining warning levels using risk matrices. Unlike some other warning systems based on risk matrices, our framework is coherent in the sense that the warning level cannot decrease if certainty and/or severity increases. This is achieved by requiring that the severity categories that represent warning conditions are well-defined and *nested*, and that forecasters select a well-defined certainty category for each of these severity categories. We also present an extension to the framework that incorporates lead time to hazard onset into warning decisions. This provides greater flexibility in warning messaging and ensures compatibility with the Common Alerting Protocol (CAP) (OASIS OPEN, 2024).

The forecast directive for this framework is to fill out the risk matrix in-line with the forecaster's probabilistic assessment of the hazard. We provide an accompanying evaluation framework that consists of a family of scoring functions that are consistent with the forecast directive. This evaluation framework encourages forecasters to follow the forecast directive, in the sense that their expected score is optimised by doing so. It also rewards forecasters who make superior probabilistic assessments. This promotes reliable warning decisions and improved hazard prediction. These consistent scoring functions are an extension of a special case of the FIRM score.

Our approach is hazard agnostic, meaning that it can be applied to any hazard or combination of hazards, and can be extended to an impact-based service if desired and appropriate data is available. The framework and evaluation method applies equally well if the labels 'warning level', 'severity' and 'certainty' are respectively replaced with 'risk level', 'impact' (or 'consequence') and 'likelihood' (or 'probability'). It is also suitable for other areas of risk management, provided that the risk matrix itself represents an adequate model of risk for the application.

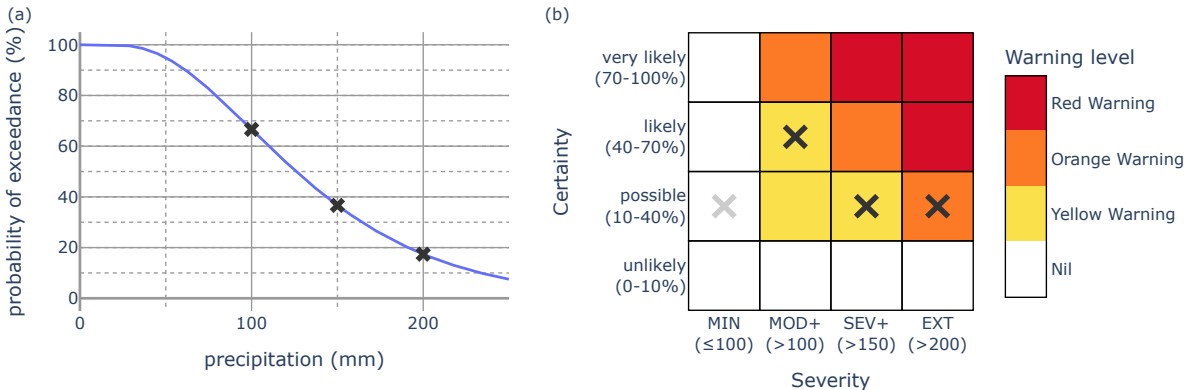

**Figure 2.** Example of a probabilistic forecast (left) for precipitation and its mapping (via dark grey crosses) to a severity–certainty matrix (right). The light grey cross indicates the certainty category for the MIN severity category implied by the probabilistic forecast. Severity thresholds, shown below the matrix, are in millimetres. In this case an Orange Warning would be issued.

The paper is structured as follows. Section 2 presents the general warning framework, illustrated by a simple example for a heavy rainfall warning service. Section 3 describes the new consistent scoring method for warnings based on our general framework. Section 4 presents a case study illustrating the warning framework and verification approach using precipitation forecasts for Tropical Cyclone Jasper. Finally, Section 5 presents a summary and discussion of results.

## 2   Warning framework using risk matrices

In this section we lay out the general framework for determining a warning level based on a risk-matrix assessment. Section 2.1 begins with a simple example to illustrate the main concepts, before laying out the general framework in plain language. Section 2.2 discusses key differences between our framework and the presentation of other frameworks, along with justification for our choices. Section 2.3 explains how the general framework can be extended to incorporate urgency, or warning lead time, into warning level determination. Section 2.4 then gives a mathematical description of the framework.

### 2.1   Illustrative example and general framework

Consider a service that warns for heavy rainfall over a 24-hour duration for Sydney, Australia. Figure 2b shows the risk matrix for this warning service. It is a 4 by 4 matrix with four certainty categories (unlikely, possible, likely and very likely) and four severity categories (MIN, MOD+, SEV+ and EXT, which correspond to severity outcomes which are negligible to minor, moderate or greater, severe or greater, or extreme). Each of the categories is clearly defined. An outcome is MOD+ (respectively SEV+ and EXT) if the 24-hour rainfall exceeds 100 mm (respectively 150 mm and 200 mm), while an outcome is MIN if the 24-hour rainfall does not exceed 100 mm. A probability $p$ is in the unlikely, possible, likely and very likely categories if

respectively $0\% \leq p < 10\%$, $10\% \leq p < 40\%$, $40\% \leq p < 70\%$ and $70\% \leq p \leq 100\%$. The associated probability thresholds, or probability decision points, are $p_1 = 10\%$, $p_2 = 40\%$ and $p_3 = 70\%$.

The warning service has four warning levels: Nil, Yellow Warning, Orange Warning and Red Warning. These are listed in order of increasing threat and each level is paired with a colour (white, yellow, orange and red) of corresponding increasing intensity. Cells in the matrix are coloured to indicate the warning scaling, which is used to determine the level at which a warning is issued.

The *forecast directive* requests that the forecaster selects, for each severity category, the unique certainty category that
aligns with their probabilistic assessment. By doing so they will select one cell in each column of the risk matrix. The *warning directive* requests that the forecaster follows the forecast directive and then chooses the warning level that is the highest warning level of the selected cells in the matrix.

For example, suppose that a forecaster has a predictive distribution for 24-hour rainfall shown by the probability of exceedance (PoE) curve in Fig. 2a. The forecaster's assessment is that the probability of exceeding 100 mm (i.e., observing a
MOD+ outcome) is $66\%$, which lies in the 'likely' category. If the forecaster follows the forecast directive, they will select the 'likely' cell in the MOD+ column of the matrix. The dark grey crosses on Fig. 2 show the probabilistic assessment of the forecaster and how these are mapped across to the matrix by following the forecast directive. The highest warning level from the selected cells is orange, and therefore by following the warning directive an Orange Warning will be issued.

The example above illustrates most of the key ingredients of this warning framework. These are: the outcome space, the
severity and certainty categories, the probability thresholds, the warning levels, the warning scaling, a forecast for the warning service, the forecast directive, the warning directive and the evaluation weights. Each of these will be described in more general terms below, with evaluation weights deferred to Section 3.

The *outcome space* is the set of all possible outcomes, which in the Sydney rainfall example would be all possible rainfall totals (in millimetres). The *severity categories* are categories of increasing hazard that together cover every possible outcome.
The severity categories, excluding the first, are *nested*, so that in the Sydney rainfall example the set of outcomes in MOD+ includes those in SEV+, which in turn includes those in EXT. The first severity category (MIN) consists of all remaining possible outcomes, covering all events for which no warning would ever be issued. The *certainty categories* form a partition of probability space into categories of increasing likelihood, with corresponding *probability thresholds*. The *warning levels* are ordered from lowest threat to highest threat.

The *warning scaling* maps each cell in the risk matrix to a warning level. We adopt the standard risk matrix convention that severity categories (the columns) increase from left to right and certainty categories (the rows) increase from bottom to top. The warning scaling has three key properties: (a) the left column and bottom row are assigned the lowest warning level (Nil), (b) warning levels are non-decreasing moving from left to right along each certainty row and (c) warning levels are non-decreasing moving from bottom to top along each severity column. Property (a) ensures that the warning service is not in a perpetual state
of warnings while properties (b) and (c) ensure that any increase of the warning level (given a fixed warning scaling) is due to either increased likelihood of the hazard, increased severity of the hazard, or both.

A *forecast* for the warning service is the selection of one certainty category for each severity category (i.e., a cross in each column of the risk matrix). The *forecast directive* and *warning directive* have already been described. However, selecting a cell in the first column of the matrix will not affect the warning level (because the first column always gives the lowest warning level of Nil), so both directives will henceforth only require a categorical certainty forecast for all severity categories except the first. (In practice, forecasters may be encouraged to make a forecast in the first column for communication purposes). While a forecaster needn't follow the directives, one major contribution of this paper is to present an evaluation framework that encourages forecasters to do so.

Finally, note that the framework presented here is quite general. The number of severity categories, certainty categories and warning levels could be different from that illustrated with the Sydney rainfall example, as could the labelling of each of the categories or levels. It is also able to handle a variety of situations, including phenomena where hazards occur at either end of a spectrum and warnings for multiple hazardous phenomena. For example:

(E1) A warning service for hazardous outdoor temperatures (either cold or hot) could define a MOD+ severity category as temperatures greater than $35°C$ or less than $0°C$, a SEV+ severity category as temperatures greater than $40°C$ or less than $-5°C$, and a EXT category as temperatures greater than $45°C$ or less than $-10°C$.

(E2) A warning service for large hailstones and/or damaging wind gusts in a 24-hour period could define a MOD+ category as events where the maximum hailstone diameter is at least 20 mm and/or the maximum wind gust is at least 25 ms$^{-1}$, and a SEV category where the maximum hailstone diameter is at least 50 mm and/or the maximum wind gust is at least 35 ms$^{-1}$. There is no EXT category in this example.

More generally, the framework could be applied to an index, which itself represents complex multi-hazard interactions. An example of such an index is the Fire Behaviour Index (FBI) used in the Australian Fire Danger Ratings System (AFDRS), which combines weather and fuel state information to determine the severity of fire behaviour (Hollis et al., 2024). Although we call the categories associated with columns 'severity categories', our general framework is also applicable to impact–likelihood matrices, where the outcome space is the set of possible impacts from the phenomena of concern.

## 2.2 Comparison with other warning schemes based on risk matrices

Several features of the warning framework merit further discussion as they differ structurally from some risk matrix warning tools proposed elsewhere or used by operational centres. A key comparison is with the work of the UKMO, to which the current authors are indebted. Although some of the labels are different (e.g., the UKMO warn for impact rather than severity) the essential *structure* of their matrix-based approach, shown in Fig. 1, can be translated to our representation in a straightforward manner. This is demonstrated in Fig. 3, where the matrix marked (a) conforms to our framework, while the matrix marked (b) is a translation of the UKMO matrix. Other authors or organisations present risk matrices using a structure with similarities to that of the UKMO (e.g., National Aeronautics and Space Administration (NASA) (2011, Fig. 53), Aldridge et al. (2020, Figs. 1 and 10)).

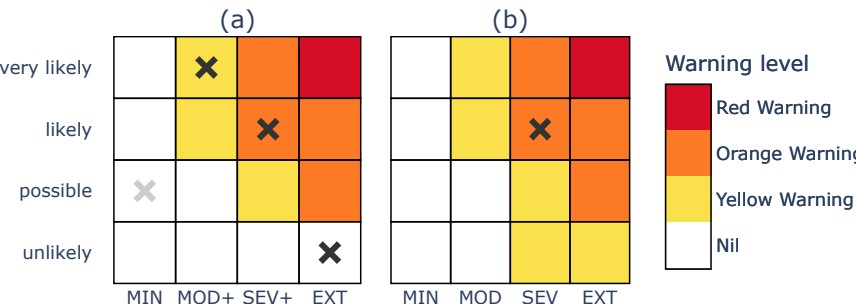

**Figure 3.** Two matrices demonstrating key structural differences between the framework outlined in Section 2.1, indicated by (a), and the approach used by the UKMO, indicated by (b).

There are three major structural differences between the UKMO warning framework, as presented in their risk matrix assessments, and the general framework detailed in Section 2.1. First, the UKMO matrix assigns a Yellow Warning level to some cells in the bottom row of their matrices, contravening property (a) for the warning scaling. The implication, if taken literally, would be a perpetual warning at the Yellow Warning level or higher, because the risk of severe weather is at least 'unlikely'. Indeed, Neal et al. (2014) define the lowest certainty category to be the set of probabilities less than 20% and do not prescribe a lower bound above 0% for this category.

The second difference is that when each of the severity categories of the UKMO matrices are labelled (e.g., Neal et al. (2014, Fig. 8)), they appear to be a partition of the outcome space, akin to (MIN, MOD, SEV, EXT), rather than a predominantly nested sequence of severity categories (MIN, MOD+, SEV+, EXT). To understand why we advocate the nesting property, consider the following forecasts for the matrix of Fig. 2b, but altered so that the severity categories are MIN, MOD, SEV and EXT:

- Forecast A: the forecast probability that the outcome is MOD, SEV and EXT is respectively 9%, 9% and 9%. The resulting warning level is a Nil warning.

- Forecast B: the forecast probability that the outcome is MOD, SEV and EXT is respectively 11%, 0% and 0%. The resulting warning level is a Yellow warning.

We argue that Forecast A is the more threatening: in Forecast A the probability that the outcome is MOD+ is 27%, while in Forecast B the probability that the outcome is MOD+ is only 11%. However, because the severity categories are MIN, MOD, SEV and EXT (rather than MIN, MOD+, SEV+ and EXT), Forecast A results in a lower warning level than Forecast B.

The third difference is that the framework presented here requires a certainty forecast for *each* of the nested severity categories (MOD+, SEV+, EXT), so that the full scope of potential threat is brought to bear on the final warning level issued. On the other hand, a forecast for each column is not a requirement for the warning service of the UKMO, as illustrated by the single 'tick' in Fig. 1 and single cross in Fig. 3b.

Nonetheless, our framework and the UKMO warning service framework, *as actually practiced*, may have fewer differences than mentioned above. For example, perpetual warnings are not a feature of their service because some small positive likelihood

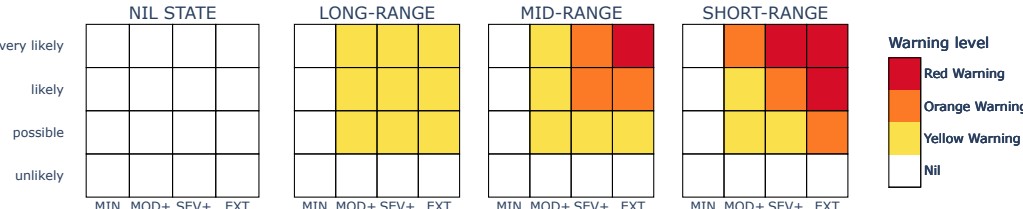

**Figure 4.** Examples of three scaling matrices and the trivial 'NIL STATE' matrix.

of a hazard is required to issue a warning, notwithstanding the presentation of their risk matrices. Similarly, the ensemble-based first-guess support tool for the warning service (MOGREPS-W) has a positive probability as its lower bound for generating a warning, as well as meteorological exceedance criteria for impact categories suggestive of the nesting property (Neal et al., 2014).

### 2.3 Warnings that vary with lead time

In many global alerting systems, the warning level is informed by the urgency of the hazard (Neußner, 2021), which typically depends on the time to onset. To accommodate this, warning scaling can be defined to vary with lead time if desired, allowing warning level to change as time to hazard onset reduces, even if the severity and certainty of the threat remain constant. This approach enables greater control of the communication and dissemination of information to the public, as discussed further in Section 5.

To incorporate lead time into warning level decisions, the composite warning scaling can be thought of as a series of risk matrix 'phases' which apply during distinct time intervals in the warning period. This is illustrated in Fig. 4, which shows three sequential matrices applying at 'LONG-RANGE', 'MID-RANGE' and 'SHORT-RANGE', in addition to the trivial 'NIL STATE' matrix. In this example, for a given position in the matrix, warning level is non-decreasing with reducing lead time. That is, for any constant severity–certainty pair, the warning level remains constant or increases as the hazard becomes more imminent. In all cases, the longest lead time phase is the trivial 'NIL STATE' matrix in which the lowest warning level is applied for all severity and certainty couples. Any standard two-dimensional warning scaling can be viewed as special case of this approach with two lead time phases; the shorter lead time phase consisting of the specified scaling matrix, and the longer lead time phase being the 'NIL STATE' matrix.

### 2.4 Mathematical description of the general framework

Each of the elements of the warning framework described in Section 2.1 has a corresponding mathematical description, presented in Table 1. Readers who prefer to avoid mathematical details may skip ahead to Section 3. For readers who would to like understand how risk matrix forecasts and the warnings generated by them can be evaluated using consistent scoring functions, the most important aspect of Table 1 is the following notation: for appropriate index $i$, each severity category is denoted by $S_i$, each certainty category by $C_i$, each probability threshold by $p_i$ and each warning level by $\ell_i$.

**Table 1.** A mathematical description of the elements of the warning framework. The integers $m$, $n$ and $q$ are positive. Evaluation weights, the final entry of the table, will be discussed in Section 3.

| element | mathematical description |
| --- | --- |
| outcome space $\Omega$ | The set of all possible realisations or possible outcomes, some of which are hazardous. |
| partition $(\Omega_0,\ldots,\Omega_m)$ of $\Omega$ | A partition of $\Omega$ ordered by increasing hazard. |
| severity categories $(S_0,\ldots S_m)$ | Ordered subsets of $\Omega$ of increasing hazard given by $S_0 = \Omega_0$ and $S_i = \cup_{j\geq i}\Omega_j$ whenever $1 \leq i \leq m$. These have the property that $S_m \subset S_{m-1} \subset \ldots \subset S_1 = S_0^c$. The set of severity categories is denoted $\mathcal{S}$. |
| certainty categories $(C_0,\ldots,C_n)$ | A partition of probability space $[0,1]$ into categories $C_i$ of increasing likelihood. The set of certainty categories is denoted $\mathcal{C}$. |
| probability thresholds $(p_1,\ldots,p_n)$ | Thresholds that partition $[0,1]$ into the certainty categories. We adopt the convention that $C_i = [p_i, p_{i+1})$ whenever $1 \leq i \leq n-1$. |
| warning levels $(\ell_0,\ldots,\ell_q)$ | A tuple of all possible warning states. The set of warning levels is equiped with a total order $\prec$ such that $\ell_0 \prec \ell_1 \prec \ldots \prec \ell_q$, where $\ell_0$ represents the state of no warning. The set of warning levels is denoted $\mathcal{L}$. |
| warning scaling $T$ | A mapping $T : \mathcal{S} \times \mathcal{C} \to \mathcal{L}$ with the following properties:<br>(a) $T(S_i,C_0) = T(S_0,C_j) = \ell_0$ for any $i$ and $j$<br>(b) $T(S_k,C_i) \preceq T(S_k,C_j)$ whenever $i \leq j$ and for any $k$<br>(c) $T(S_i,C_k) \preceq T(S_j,C_k)$ whenever $i \leq j$ and for any $k$. |
| forecast $F$ for the warning service | A tuple of the form $F = (f_1,\ldots,f_m)$, where each $f_i$ is the forecast certainty category for the severity category $S_i$. The set $\mathcal{F}$ of all internally consistent forecasts is given by<br>$$\mathcal{F} = \{(f_1,\ldots,f_m) \in \mathcal{C}^m : \text{the sequence } f_1,\ldots,f_m \text{ is non-increasing}\}.$$ |
| the forecast directive | Issue the forecast $F = (f_1,\ldots,f_m)$ such that $f_i \in \mathcal{C}$ satisfies $\mathbb{P}(S_i) \in f_i$. Here $\mathbb{P}(S_i)$ is the forecaster's probabilistic assessment of the outcome belonging to $S_i$. |
| the warning directive | Follow the forecast directive with issued forecast $F$, then issue the warning level $\max\{T(S_i,f_i) : 1 \leq i \leq m\}$. |
| evaluation weights $(v_1,\ldots,v_q)$ | A tuple of positive weights used for evaluating the accuracy of forecasts. The weight $v_i$ is proportional to the importance of accurately discriminating warning states below level $\ell_i$ from states at or above level $\ell_i$. |

An interval of the real numbers $\mathbb{R}$ with endpoints $a$ and $b$ will be denoted by $[a,b]$ if both endpoints are included in the interval, $[a,b)$ if only $a$ is included and $(a,b)$ if both endpoints are excluded. The convention that the left endpoint of $C_i$ is included but not the right endpoint, as per $C_i = [p_i, p_{i+1})$, can be changed, with appropriate modification to the scoring functions introduced in Section 3.

To illustrate some of the notation of Table 1, examples (E1) and (E2) of Section 2.1 can be stated as follows.

(E1) This warning service for hazardous temperatures has outcome space $\Omega = (-273.15, \infty)$ in units of degrees Celsius. Its severity categories are defined by $S_0 = [0, 35]$, $S_1 = (-273.15, 0) \cup (35, \infty)$, $S_2 = (-273.15, -5) \cup (40, \infty)$ and $S_3 = (-273.15, -10) \cup (45, \infty)$.

(E2) This warning service for hazardous hailstones and wind gusts has outcome space $\Omega = [0,\infty) \times [0,\infty)$, where a possible outcome $(h,g)$ in $\Omega$ is a combination of the maximum hailstone diameter $h$ in millimeters and the maximum wind gust $g$ in ms$^{-1}$. Its severity categories are defined by $S_0 = [0,20) \times [0,25)$, $S_1 = [20,\infty) \times [25,\infty)$ and $S_2 = [50,\infty) \times [35,\infty)$.

If the matrix scaling also depends on lead time, then warning phase is an additional element required in the framework. In this case, the warning scaling $T$ is also a function of warning phase with the additional property that the warning level does not decrease with decreasing lead time, all else being equal.

## 3 Consistent scoring of warnings to evaluate warning systems

It is important to know how well competing forecasters or forecast procedures perform when generating warning products based on risk matrices. For example, one may want to know whether an upgrade to a forecast system produces more accurate warnings, or whether the current method of producing warnings performs better than not issuing a warning at all (i.e., always issuing a 'Nil' warning level).

In this section, we present a new scoring function, called the *risk matrix score*, that enables such questions to be answered. The risk matrix score gives a measure of forecast error, expressed as a number, for a particular forecast case when compared to the corresponding observation. The smaller the score, the better the forecast. A perfect score is 0. Forecasters or forecasting procedures can be compared by calculating the mean risk matrix score over many forecast cases, with a smaller mean risk matrix score indicating better predictive performance on average. The risk matrix score is also a *consistent* scoring function, which means that a forecaster's expected score will be minimised (i.e., optimised) by following the forecast directive. This property implies that the risk matrix score gives lower (i.e., better) scores to forecasters who make accurate probabilistic assessments and follow the forecast directive.

We also introduce a special case of the risk matrix score, called the *warning score*, which assesses predictive accuracy of the final warning level. It does this by measuring forecast error only for those decision points in the risk matrix that are relevant to warning level determination. To calculate the warning score, the warning service must specify the relative importance of each warning level decision point. This specification is called the *evaluation weights*.

The main ideas are introduced, in a non-technical manner, in Section 3.1 using the example of a flood warning service. The risk matrix score and warning score are properly defined in Sections 3.2 and 3.3. In Section 3.4 we present a synthetic experiment involving six fictitious forecasters who have access to different information (which affects the accuracy of their probabilistic assessments) and have different strategies when filling out the risk matrix (such as following the warning directive, over-warning or under-warning). The synthetic experiment illustrates desirable properties of the risk matrix score and warning score. Both scores correctly rank (a) forecasters who follow the directive but who differ in the accuracy of their probabilistic assessments, and (b) forecasters who have identical probabilistic assessments but who differ to the degree in which they follow the forecast directive.

The risk matrix score (and by implication, the warning score) has a Python implementation in the open source `scores` package (Leeuwenburg et al., 2024). The `scores` documentation also includes a tutorial for how to use this implementation.

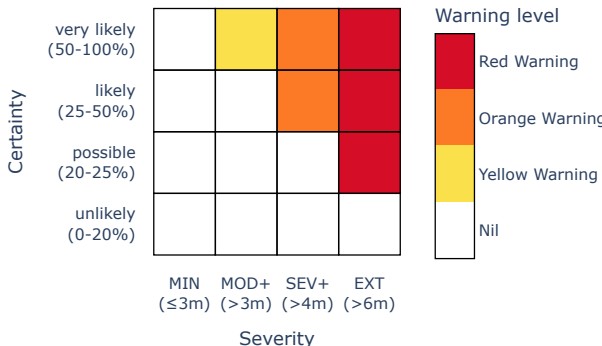

**Figure 5.** Risk matrix for a flood warning service with severity categories defined based on river height.

### 3.1 Introduction to scoring methods and evaluation weights

The basic ideas of warning service design and consistent scoring methods will be illustrated using a flood warning service for a fictitious town situated on a river. The severity categories MOD+, SEV+ and EXT are defined using river height exceedance thresholds of 3 metres, 4 metres and 6 metres respectively.

When the river height exceeds 3 meters (MOD+), the low level causeway is flooded leading to lengthy detours and traffic bottlenecks around the high level bridge. Consultation with community members in the town suggests that a miss (i.e., forecast

under 3m but observation over 3m) causes similar problems to a false alarm (i.e., forecast over 3m but observation under 3m). This means that the optimal probability threshold for warning for the MOD+ category is 50%.[1]

When the river height exceeds 4 metres (SEV+) low lying areas of the town are inundated resulting in the closure of the primary school. Community consultation suggests that a miss (i.e., forecast under 4m but observation over 4m) is about three times worse than a false alarm (i.e., forecast over 4m but observation under 4m). This means that the optimal probability

threshold for warning for the SEV+ category is 25%.

When the river height exceeds 6 metres (EXT), a river levee is breached and significant inundation occurs resulting in a third of the town becoming isolated by flood waters. Community consultation suggests that a miss (i.e., forecast under 6m but observation over 6m) is about four times worse than a false alarm (i.e., forecast over 6m but observation under 6m). This means that the optimal probability threshold for warning for the EXT category is 20%.

A Yellow Warning is warranted if the risk of exceeding the MOD+ threshold of 3 metres is at least 50%. An Orange Warning is warranted if the risk of exceeding the SEV+ threshold of 4 metres is at least 25%. A Red Warning is warranted if the risk of exceeding the EXT threshold of 6 metres is at least 25%. The warning level issued (if not Nil) is the highest such level. This flood warning service can be represented by the risk matrix depicted in Fig. 5.

---

[1]In general, if the ratio of the cost of a miss to the cost of a false alarm is $c_m : c_f$, then the optimal probability threshold for issuing a warning for the event is $c_f/(c_f + c_m)$ (Taggart et al., 2022). In this case, we have $c_m : c_f = 1 : 1$ and hence the selected probability threshold is $c_f/(c_f + c_m) = 1/2 = 50\%$.

We now discuss how forecasts for this warning service are scored in a manner that is consistent for this warning service. To follow the forecast directive, the forecaster must select one cell (i.e., certainty category) for each of the MOD+, SEV+ and EXT columns in the risk matrix. For each column, there are three probability decision points corresponding to the probability thresholds 20%, 25% and 50%. In total, there are nine decisions points (three probability thresholds for each of the three columns). For example, for the (EXT, 20%) decision point, the forecaster must decide whether the likelihood of an EXT outcome is less than 20% or at least 20%.

To construct the risk matrix score we first score, in a consistent manner, the forecast for each of the nine decision points. Consistency here means that the scores for a miss and a false alarm are appropriate for the particular probability threshold. For example, for the (EXT, 20%) decision point:

- a false alarm (forecast is at least 20% and EXT is not observed) is given a score of 0.2

- a miss (forecast is less than 20% and EXT is observed) is given a score of 0.8

- a hit (forecast is at least 20% and EXT is observed) is given a score of 0

- a correct negative (forecast is less than 20% and EXT is not observed) is given a score of 0.

Note that scores are oriented so that a lower score implies a better forecast. For the (EXT, 20%) decision point, a miss is penalised 4 times more than a false alarm, which is consistent with community consultation. The score for a false alarm is equal to the probability threshold for this decision point (i.e., 20% or 0.2), while the score for a miss is equal to the complementary probability (i.e., 80% or 0.8). Hits and correct negative receive a perfect score of 0.

The risk matrix score is then a *weighted* sum of all the scores for each of the nine decision points. If all decision points were of equal importance for the warning service, it would be appropriate to apply a weight of 1 for each of the nine decision points. In general, weights should be non-negative, and if we were to assess how forecasters fill out the risk matrix (without regard to warning scaling or warning level), then we should take positive weights for each decision point.

However, for this flood warning service, not every decision point is equally important. In fact, for determining the final warning level, there are exactly three relevant decision points: (MOD+, 50%), (SEV+, 25%) and (EXT, 20%). That is, to determine the final warning level, we only need to know whether the forecaster forecast above or below each of those decision points. If the accuracy of the final warning level is all that matters, then we put positive weights on the scores for those three decision points and assign a weight of zero for the remaining six decision points.

This segues into the concept of *evaluation weights*. Evaluation weights quantify the importance of accurately discriminating between each of the warning levels. For this flood warning service, the Nil warning versus Yellow Warning decision point is the least important. The costs of preemptively erecting detour signs at the low level causeway if the river stays below three metres, or of belatedly reacting to the situation after the causeway has flooded, is low compared to the corresponding costs at the Yellow warning versus Orange Warning decision point. At the Yellow–Orange Warning level decision point, an incorrect forecast results in either unnecessary preemptive school closures or reactive school closures. At the Orange warning versus Red Warning decision point, the stakes of an incorrect warning decision are higher still, resulting in either unnecessary evacuations

or stranded communities. Community consultation suggests that the relative importance of the Nil–Yellow, Yellow–Orange and Orange–Red warning level decision points are in the ratio 1:3:10. We say that the *evaluation weights* for this flood warning service are $(1, 3, 10)$.

This gives rise to the *warning score*, which is a special case of the risk matrix score, but with weights in the weighted sum determined by the warning scaling and evaluation weights (see Section 3.3 for details). For the flood warning service, this means that the warning score is the weighted sum of the scores for each of the decision points in the risk matrix, where the weight on the (MOD+, 50%) decision point is 1, the weight on the (SEV+, 25%) decision point is 3, the weight on the (EXT, 20%) decision point is 10, and the weight on the six remaining decision points is 0.

Evaluation weights should be given to forecasters in advance of them issuing warnings. The weights indicate the relative importance of each decision point and indicates how forecasters should allocate resources, given time and information constraints, to make the best possible warning decisions. Although the process for determining weights in this fictitious flood example was presented straightforwardly, this framework motivates further research into developing best practices for eliciting thresholds and weights through stakeholder consultation.

## 3.2    The risk matrix score

In this section we give a mathematical definition of the risk matrix score and advocate its usage from the perspective of decision theory.

     There are many different scoring functions that could be used for evaluating forecasts for risk matrices, but not all reward desirable, or penalise unwanted, forecasting behaviour. Powerful arguments have been presented (Murphy and Daan, 1985;

Engelberg et al., 2009; Gneiting, 2011) that when the requested forecast is not in the form of a full probability distribution, either the scoring function should be made known to the forecaster ahead of time or a 'forecast directive' should be given to the forecaster for how to convert their probability distribution to a forecast in the requested form. In this second case, the scoring function should be *consistent* with the forecast directive, which essentially means that the forecaster's expected score will be optimised by following the forecast directive.

We agree with this perspective. In our case, the forecast directive has been specified (see Section 2.1 and Table 1), so in the following we construct a family of scoring functions that are consistent with the forecast directive. See Appendix A for a precise mathematical definition of consistency in this context. As illustrated in Section 3.4, using a consistent scoring function provides the following benefits: (i) it rewards forecasters who make accurate probability assessments and follow the forecast directive, (ii) it penalises forecasters who follow the forecast directive but use poorer probabilistic assessments and (iii) it

penalises forecasters who make accurate probabilistic assessments but do not follow the forecast directive (Holzmann and Eulert, 2014).

     We begin by using so-called elementary scoring functions (Ehm et al., 2016) to score the forecast relative to each decision point in the risk matrix. Using the notation of Table 1, the elementary scoring function $\mathrm{ES}_{i,j}$ associated severity category $S_i$

and probability threshold $p_j$ is given by

$$\text{ES}_{i,j}(F,y) = \begin{cases} p_j, & y \notin S_i \text{ and } f_i \text{ lies on or above } p_j \\ 1-p_j, & y \in S_i \text{ and } f_i \text{ lies below } p_j \\ 0, & \text{otherwise,} \end{cases} \tag{1}$$

where $F = (f_1, \ldots, f_m)$ is the forecast for the warning service, $f_i$ is the corresponding forecast for the severity category $S_i$ and $y$ is the observation in the outcome space $\Omega$. In other words, $\text{ES}_{i,j}$ gives a score for forecasts relative to the $(S_i, p_j)$ decision point in the risk matrix, giving a penalty of $p_j$ for a false alarm, a penalty of $1-p_j$ for a miss and a perfect score of 0 otherwise.

The *risk matrix scoring function* RMaS is defined by

$$\text{RMaS}(F,y) = \sum_{i=1}^{m} \sum_{j=1}^{n} w_{i,j} \text{ES}_{i,j}(F,y), \tag{2}$$

where $F$ is the forecast for the warning service and $y$ is the observation in the outcome space $\Omega$, and each $w_{i,j}$ is a pre-specified non-negative weight with at least one such weight strictly positive. In other words, the risk matrix score is a weighted sum of the forecast errors associated with each of the decision points $(S_i, p_j)$ of the risk matrix. The weight $w_{i,j}$ indicates the importance of issuing a forecast on the same side as the observation relative to the $(S_i, p_j)$ decision point; failure to do so results in penalties for misses and false alarms amplified by a factor of $w_{i,j}$. Hence the choice of weights should reflect which decision points are most important for the warning service. If all decision points have equal importance then each weight could be assigned a value of 1. From the forecaster's perspective, the choice of weights guide where to focus improvements in forecast system development.

The proof that the risk matrix score is consistent with the forecast directive is given in Appendix A. In essence, it is shown that each elementary scoring function is consistent and that consistency is preserved through weighted sums of consistent scoring functions. The choice of non-negative weights does not affect the score's consistency with the forecast directive nor the optimal conversion rule from a forecaster's full probability distribution to the forecast for the warning service. However, the weights do indicate which decision points may require more attention when fine-tuning forecasting procedures.

The risk matrix score can be decomposed as a sum of contributing scores for the forecast for each column of the risk matrix, excluding the leftmost column. That is, $\text{RMaS}(F,y) = \sum_{i=1}^{m} \text{CS}_i(F,y)$, where $\text{CS}_i(F,y) = \sum_{j=1}^{n} \text{ES}_{i,j}(F,y)$. We call $\text{CS}_i(F,y)$ the *column score* associated with severity category $S_i$. The column score for $S_i$ is presented in tabular form in Table 2 for the case when $n = 3$ (i.e., there are three probability thresholds). Note that a perfect column score of 0 is achieved when the lowest certainty category $C_0$ was forecast and the observation $y$ was not in $S_i$, or when the highest certainty category $C_3$ was forecast and the observation $y$ was in $S_i$.

A tabular form of the risk matrix score is a concatenation of tables of column scores. Table 3 illustrates this for the Sydney rainfall example (Fig. 2b), where we have set each weight $w_{i,j}$ to be 1. Note that on average, false alarms (non-zero scores in the left column of each severity category) are penalised less than misses (non-zero scores in the right column of each severity category). This is because the probability thresholds on average are closer to 0 than to 1.

**Table 2.** The column score associated with severity category $S_i$ is given by the entries in the table for the case where there are four certainty categories $C_0$ to $C_3$. The score depends on whether the observation $y$ belongs to $S_i$ and which certainty category was forecast.

| forecast | $y$ not in $S_i$ | $y$ in $S_i$ |
|---|---|---|
| $C_3$ | $p_3 w_{i,3} + p_2 w_{i,2} + p_1 w_{i,1}$ | $0$ |
| $C_2$ | $p_2 w_{i,2} + p_1 w_{i,1}$ | $(1-p_3)w_{i,3}$ |
| $C_1$ | $p_1 w_{i,1}$ | $(1-p_2)w_{i,2} + (1-p_3)w_{i,3}$ |
| $C_0$ | $0$ | $(1-p_1)w_{i,1} + (1-p_2)w_{i,2} + (1-p_3)w_{i,3}$ |

**Table 3.** Components of the risk matrix score in tabular form for the Sydney rainfall example, where each of the weights $w_{i,j}$ is 1.

| forecast | obs not in MOD+ | obs in MOD+ | obs not in SEV+ | obs in SEV+ | obs not in EXT | obs in EXT |
|---|---|---|---|---|---|---|
| very likely | 1.2 | 0 | 1.2 | 0 | 1.2 | 0 |
| likely | 0.5 | 0.3 | 0.5 | 0.3 | 0.5 | 0.3 |
| possible | 0.1 | 0.9 | 0.1 | 0.9 | 0.1 | 0.9 |
| unlikely | 0 | 1.8 | 0 | 1.8 | 0 | 1.8 |

We now calculate the risk matrix score for the forecast in the Sydney rainfall example (Fig. 2b), again assuming that each of the weights $w_{i,j}$ is 1. The observed rainfall was 136 mm, which lies in MOD+ but not in SEV+ nor EXT. The MOD+ forecast was 'likely' so the column score for the MOD+ category is 0.3. The SEV+ and EXT forecasts were both 'possible', so the column score for SEV+ and EXT are both 0.1. The risk matrix score is the sum of these contributions, namely 0.5. Had another forecaster failed to issue a warning by forecasting 'unlikely' for MOD+, SEV+ and EXT, they would have received a risk matrix score of 1.8. With this choice of weights, the second forecaster performed worse than the first forecaster because they received a higher risk matrix score.

The risk matrix score is an extension of two special cases of the FIRM score. The risk matrix score with exactly two certainty categories reduces to one case of the FIRM score (Taggart et al., 2022, Section 4.1), while the risk matrix score with exactly two severity categories reduces to another case of the FIRM score (Taggart et al., 2022, Section 5.4). Moreover, our risk matrix warning framework and accompanying risk matrix score can also handle situations beyond the FIRM framework. This includes warning services with levels whose the probability thresholds decrease with increasing severity (e.g., Fig. 5), thereby providing a coherent solution to problems discussed by Taggart et al. (2022, Appendix E). On the other hand, the FIRM framework provides a decision-theoretically coherent framework for (a) reducing the penalties applied to near misses and close false alarms and (b) warnings based on expected value forecasts. Incorporating these features into a generalisation of the risk matrix score is one avenue for future research.

Finally, although not formally proved in this article, the characterisation of consistent scoring functions for the expectation functional (Savage, 1971; Gneiting, 2011; Ehm et al., 2016), applied to the context of binary outcomes, essentially implies that any scoring function that is consistent with the forecast directive is in the form of the risk matrix score. Measures of risk that are not of this form (e.g., the product of certainty and severity) will be inconsistent with the forecast directive and hence

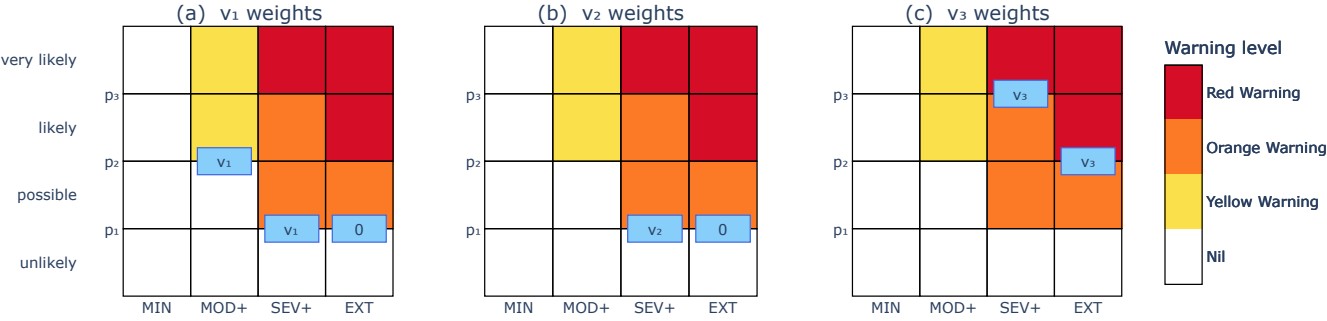

**Figure 6.** Illustration of the algorithm for weight selection using the ZETA scaling.

with decision-theoretic coherent use of risk matrices. Although not expressed in these terms, this is the essence of a major criticism of the conventional use of risk matrices in risk management (Cox, 2008). Our framework is suitable for any area of risk management provided that the risk matrix itself represents an adequate model of risk for the application.

### 3.3 The warning score

The warning score WS is a special case of the risk matrix score RMaS, where the weights $w_{i,j}$ are selected to assess predictive performance only for those decision points $(S_i, p_j)$ in the risk matrix that potentially lead to a change in the final warning level. Decision points $(S_i, p_j)$ that are irrelevant for warning level determination have corresponding weights $w_{i,j}$ set to 0. The classification of a weight being either zero or positive is determined by the warning scaling. The value of each positive weight is determined by pre-specified *evaluation weights*, as detailed below.

To appreciate the classification of weights as a function of warning scaling, consider the LONG-RANGE scaling of Fig. 4 as a simple example. In the MOD+ column, the only important decision point, as far as warning level is concerned, is the $p_1$ probability threshold (that is, the threshold between 'unlikely' and 'possible'), because a change in warning level occurs at that threshold. A forecaster only need decide whether MOD+ is (a) 'unlikely' or (b) 'possible' or higher to determine the forecast warning level for the column. Moreover, because of the nesting property for severity categories, the forecast relative to the (MOD+, $p_1$) decision point completely determines the warning level. Hence the weights for the warning score WS for the LONG-RANGE scaling are all 0 except for the positive weight $w_{1,1}$, which corresponds to the sole decision point (MOD+, $p_1$) that determines the final warning level.

Recall from Table 1 that $(\ell_0, \ldots, \ell_q)$ is the tuple of all possible warning levels. To determine the choice of values for the positive weights in the warning score, we introduce corresponding warning *evaluation weights* $(v_1, \ldots, v_q)$. Each weight $v_i$ is a positive number that is proportional to the importance of accurately discriminating warning states below the level $\ell_i$ from warning states at or above the level $\ell_i$. See the flood warning service of Section 3.1 for an example of how evaluation weights might be determined.

**Table 4.** Weights $w_{i,j}$ associated with each $(S_i, p_j)$ decision point for the ZETA, LONG-RANGE, MID-RANGE and SHORT-RANGE scalings when the evaluation weights $(v_1, v_2, v_3)$ are used. Here the severity categories are given by $S_1 = $ MOD+, $S_2 = $ SEV+ and $S_3 = $ EXT.

|       | ZETA |         |       | LONG-RANGE |       |       | MID-RANGE |       |       | SHORT-RANGE |       |       |
|-------|------|---------|-------|------------|-------|-------|-----------|-------|-------|-------------|-------|-------|
| $p_3$ | 0    | $v_3$   | 0     | 0          | 0     | 0     | 0         | 0     | $v_3$ | $v_2$       | $v_3$ | 0     |
| $p_2$ | $v_1$| 0       | $v_3$ | 0          | 0     | 0     | 0         | $v_2$ | 0     | 0           | $v_2$ | $v_3$ |
| $p_1$ | 0    | $v_1+v_2$ | 0   | $v_1$      | 0     | 0     | $v_1$     | 0     | 0     | $v_1$       | 0     | $v_2$ |
|       | $S_1$| $S_2$   | $S_3$ | $S_1$      | $S_2$ | $S_3$ | $S_1$     | $S_2$ | $S_3$ | $S_1$       | $S_2$ | $S_3$ |

Given a warning scaling and evaluation weights $(v_1, \ldots, v_q)$, the weights $w_{i,j}$ for the corresponding warning score WS can be determined by an algorithm that identifies which decision points are critical for determining the final warning level. The algorithm is stated and justified in general terms in Appendix B. Here we illustrate the algorithm using evaluation weights $(v_1, v_2, v_3)$ and the warning scaling of Fig. 6, which we call the ZETA scaling. The ZETA scaling illustrates all facets of the algorithm. Start with the first warning score weight $v_1$, which corresponds to discriminating Nil from Yellow Warning (or above) levels. For each column, identify any probability thresholds where the colour changes from white to yellow (or higher). These occur at the decision points (MOD+, $p_2$), (SEV+, $p_1$) and (EXT, $p_1$), as illustrated with blue boxes on Fig. 6a. For each probability threshold $p_i$ with a blue box, assign a value of $v_1$ to the left-most box, and a value of 0 to any remaining boxes at the same probability level. In this case, the decision points (MOD+, $p_2$) and (SEV+, $p_1$) were assigned the value $v_1$ and the decision point (EXT, $p_1$) the value 0. We now repeat this process for $v_2$, by identifying probability thresholds where the colour changes from yellow (or lower) to orange (or higher). The decision points are (SEV+, $p_1$) and (EXT, $p_1$). Both are at the same probability threshold so we assign a value of $v_2$ to the left-most box and 0 to the remaining, as indicated in Fig. 6b. Repeating the process for $v_3$, the value $v_3$ is assigned to each of the decision points (SEV+, $p_3$) and (EXT, $p_2$), since it is those decision points where the colour changes from orange (or below) to red as indicated in Fig. 6c. The final step of the process is to sum up all the values for each decision point, assigning a value of 0 to any remaining decision points without values. The set of weights for the ZETA scaling given by this algorithm is presented in Table 4, along with weights for the scalings of Fig. 4.

We calculate the warning score for the Sydney rainfall example (Fig. 2b), using illustrative evaluation weights $(v_1, v_2, v_3) = (1, 2, 3)$. In this case, the warning scaling is the SHORT-RANGE scaling, so the weights $w_{i,j}$ for the warning score WS are given by the three right-most columns of Table 4. Using these weights and column score of Table 2, we obtain the warning score in tabulated form (Table 5). To score a particular forecast case, a column score for each of MOD+, SEV+ and EXT is obtained using Table 5 and the warning score is the sum of these. Recall that the observation was in MOD+ but not the other severity categories. The forecaster in the Sydney rainfall example obtained column scores of 0.6, 0 and 0.2 to obtain a final warning score of 0.8. A second forecaster who forecasts 'unlikely' for each severity category receives a warning score of 1.5. So with this scaling and choice of evaluation weights, the first forecaster is again assessed as performing better. Note that the

**Table 5.** Components of the warning score WS in tabular form for the Sydney rainfall example (Fig. 2b), using evaluation weights $(1, 2, 3)$.

| forecast | obs not in MOD+ | obs in MOD+ | obs not in SEV+ | obs in SEV+ | obs not in EXT | obs in EXT |
|---|---|---|---|---|---|---|
| very likely | 1.5 | 0 | 2.9 | 0 | 1.4 | 0 |
| likely | 0.1 | 0.6 | 0.8 | 0.9 | 1.4 | 0 |
| possible | 0.1 | 0.6 | 0 | 2.1 | 0.2 | 1.8 |
| unlikely | 0 | 1.5 | 0 | 2.1 | 0 | 3.6 |

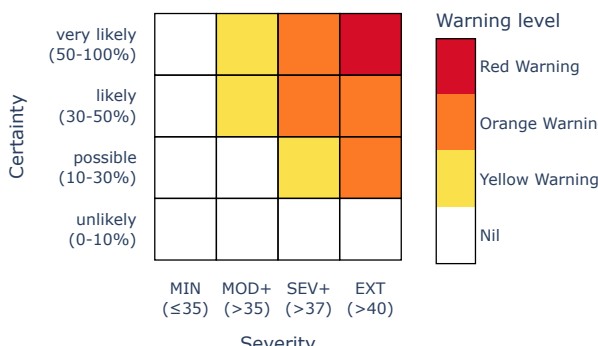

**Figure 7.** Risk matrix used for the synthetic experiment, warning for hazardous heat. The severity thresholds are for daily maximum temperatures in units of degrees Celsius.

penalties in the MOD+ columns are the same for a forecast of 'possible' or 'likely'. This is because the scaling does not change (both cells are yellow in the matrix), and this is reflected in the scoring method.

### 3.4 Case study using synthetic forecasts

We now run a simple synthetic experiment to test whether the scoring methods described above are able to correctly rank competing forecasters who have varying levels of forecast capablity and adherance to the forecast directive. We consider a simple warning service for hazardous outdoor heat, as depicted in Fig. 7, with certainty thresholds of $0.1$, $0.3$ and $0.5$, severity thresholds for daily maximum temperature of $35°C$, $37°C$ and $40°C$, and warning scaling as shown.

Let $\mathcal{N}(\mu, \sigma^2)$ denote the normal distribution with mean $\mu$ and variance $\sigma^2$. Suppose that each observed daily maximum temperature $y$ is generated from random variables $y_1$, $y_2$ and $y_3$ using the formula $y = y_1 + y_2 + y_3$, where each random variable $y_i$ is independent of the others, $y_1 \sim \mathcal{N}(20, 10^2)$, $y_2 \sim \mathcal{N}(0, 5^2)$ and $y_3 \sim \mathcal{N}(0, 2^2)$. The random variables $y_1$, $y_2$ and $y_3$ can be thought to represent variability from seasonal, synoptic scale and mesoscale weather processes respectively. The climatic distribution of maximum temperature is $\mathcal{N}(20, 10^2 + 5^2 + 2^2)$. The climatic probability of exceeding $35°C$, $37°C$ and $40°C$ on any given randomly selected day is therefore $0.093$, $0.067$ and $0.039$.

Six forecasters named NeverWarnNate, SeasonalSam, SynopticSally, RiskAverseRick, RiskTolerantReena and Playful-Pranay issue competing forecasts for this warning service. All six know the climatic distribution of maximum temperature.

- NeverWarnNate only knows about the climatic distribution. Using this distribution he follows the forecast directive to issue categorical forecasts for the warning service. This means that his categorical forecasts for MOD+, SEV+ and EXT are always 'unlikely' and that his warning level is always Nil.

- SeasonalSam has access to seasonal information $y_1$ but not synoptic or mesoscale information. She makes the ideal predictive distribution using this information, namely $\mathcal{N}(y_1, 5^2 + 2^2)$, and from this distribution follows the forecast directive to issue categorical forecasts for the warning service.

- SynopticSally has access to seasonal information $y_1$ and synoptic information $y_2$ but not mesoscale information. She makes the ideal predictive distribution using this information, namely $\mathcal{N}(y_1 + y_2, 2^2)$, and from this distribution follows the forecast directive to issue categorical forecasts for the warning service.

- RiskAverseRick has the same information as SynopticSally and makes the same ideal predictive distribution. However, he does not follow the forecast directive but instead converts his predictive distribution to categorical forecasts using lower certainty thresholds, namely 0.05, 0.2 and 0.4. Hence he warns more frequently than SynopticSally.

- RiskTolerantReena has the same information as SynopticSally and makes the same ideal predictive distribution. However, she does not follow the forecast directive but instead converts her predictive distribution to categorical forecasts using higher certainty thresholds, namely 0.2, 0.4 and 0.6. Hence she warns less frequently than SynopticSally.

- PlayfulPranay has the same information as SynopticSally and makes the same ideal predictive distribution. For each column, he identifies the warning level $\ell$ of the cell he would select to follow the forecast directive. If possible he then selects an alternative cell with warning level $\ell$, if such a cell exists, otherwise he follows the forecast directive. For example, if SynopticSally forecasts 'likely' for SEV+ then PlayfulPranay forecasts 'very likely', and if SynopticSally forecasts 'possible' for SEV+, so will PlayfulPranay. Hence PlayfulPranay issues the same warning levels as SynopticSally, but sometimes selects different certainty categories.

Based on this set-up, a good scoring function ought to rank SynopticSally ahead of SeasonalSam and SeasonalSam ahead of NeverWarnNate. All three forecasters make the best probabilistic predictions they can based on the information that is available to them, and all three follow the forecast directive. The difference between these three forecasters is that some have access to more information and can therefore make more accurate probabilistic predictions. Additionally, a good scoring function ought to rank SynopticSally ahead of RiskAverseRick and RiskTolerantReena. These three forecasters make identical probabilistic predictions of the risks. They differ because SynopticSally follows the forecast directive but RiskAverseRick and RiskTolerantReena do not. Finally, a good scoring function that takes into account every decision point in the matrix ought to rank SynopticSally ahead of PlayfulPranay, while a good scoring function that only takes into account decision points that affect warning levels should rank SynopticSally and PlayfulPranay equally.

**Table 6.** Mean risk matrix score $\overline{\mathrm{RMaS}}$ and mean warning score $\overline{\mathrm{WS}}$ for the synthetic experiment. A lower mean score is better.

| forecaster | $\overline{\mathrm{RMaS}}$ | $\overline{\mathrm{WS}}$ |
|---|---|---|
| NeverWarnNate | 0.4178 | 0.2267 |
| SeasonalSam | 0.1882 | 0.0984 |
| SynopticSally | 0.0658 | 0.0333 |
| RiskAverseRick | 0.0689 | 0.0350 |
| RiskTolerantReena | 0.0693 | 0.0352 |
| PlayfulPranay | 0.2175 | 0.0333 |

We now test the effectiveness of the risk matrix score and warning score at ranking these forecasters, where the weights for the risk matrix score RMaS and the evaluation weights are all 1. To conduct one synthetic experiment, the random variates $y_1$, $y_2$ and $y_3$ are generated, the probabilistic assessments of each forecaster computed and then their issued categorical forecasts are scored against the corresponding observation $y$. The experiment is repeated independently 1 million times, and the mean risk matrix score $\overline{\mathrm{RMaS}}$ and the mean warning score $\overline{\mathrm{WS}}$ calculated for each forecaster. The results are given in Table 6. The results show that both mean scores meet expectations of a good scoring function: SynopticSally has the lowest mean score, and SeasonalSam outperformed NeverWarnNate. The risk matrix score distinguished between SynopticSally and PlayfulPranay while the warning score did not. Standard hypothesis tests for equipredictive performance (Gneiting and Katzfuss, 2014, Section 3.3) were conducted and it was found that SynopticSally outperformed the other forecasters at the 5% significance level, with the exception of PlayfulPranay with the warning score. Mean scores from RMaS are higher than those from WS because their respective sum of weights $w_{i,j}$ is 9 and 5. Finally, even though RiskAverseRick and RiskTolerantReena do not follow the forecast directive, they still outperform SeasonalSam because the synoptic information they use gives them substantially more accurate and sharp probabilistic assessments.

## 4   Case study of warnings for heavy rainfall from Tropical Cyclone Jasper

We now illustrate the warning framework with a case study using operational ensemble precipitation forecasts for northeastern parts of Queensland, Australia, which were impacted by Tropical Cyclone Jasper during mid December 2023. The emphasis in this section is on how warning products can be generated and assessed given warning service specifications within the risk matrix framework, rather than on how those specifications are determined.

### 4.1   Specifications for warning service and resultant warning products

To begin, we describe (without justification) a simple warning service for heavy rainfall for 17 December 2023.[2] The severity categories for the warning service are defined using climatological thresholds for heavy rainfall based on annual exceedance

---

[2]Throughout this section, 17 December 2023 refers to the 24-hour period ending 17 December 2023 15:00:00 UTC. This period matches the period of 24-hour precipitation accumulation forecasts available to the authors for this study, and is offset from the local date by only one hour.

probabilities (AEPs) for a duration of 24 hours at each location. The categories MOD+, SEV+ and EXT are defined by rainfall that exceeds 20%, 10% and 5% AEPs respectively. The AEP values are taken from the 2016 Design Rainfall gridded dataset (Bureau of Meteorology, 2016; Johnson and Green, 2018) of the Australian Bureau of Meteorology (hereafter 'the Bureau'). For example, for the city of Cairns[3] the thresholds in millimetres for MOD+, SEV+ and EXT are 298 mm, 357 mm and 415 mm.

The certainty categories for the warning service consist of probabilities $p$ satisfying $0\% \leq p < 10\%$, $10\% \leq p < 30\%$, $30\% \leq p < 50\%$ and $50\% \leq p \leq 100\%$ to which we respectively assign the labels unlikely, possible, likely and very likely.

The warning scaling depends on the issuance time of the warning relative to 17 December 2023, which is the validity period of the warning. All warnings are issued a couple of hours after midnight on the local day. Warnings issued early on the day of the validity period (a 'day+0' lead time) use the SHORT-RANGE scaling (see Fig. 4). Warnings issued the day before (a 'day+1' lead time) use the MID-RANGE scaling. Warnings issued two days before the validity period (a 'day+2' lead time) use the LONG-RANGE scaling.

We compare warnings based on forecasts from three different forecast systems: the 51-member ensemble prediction system from the European Centre for Medium Weather Forecasts (ECMWF) (Molteni et al., 1996), the ECMWF's high resolution numerical weather prediction model (hereafter called ECMWF deterministic), and the Bureau's operational 18-member ACCESS-GE3 ensemble prediction system (Bureau of Meteorology, 2019). The forecasts used from these systems have an initial condition time (or base time) of 12:00 UTC (a few hours before midnight local time). These forecasts could be used to issue warning products a couple of hours after midnight local time. Gridded AEP values and forecasts from each forecast system were regridded to a common grid prior to generating warning products.

We now explain how each of these forecast systems can be used to generate probabilities for each of the severity categories MOD+, SEV+ and EXT. In the simplest case, ECMWF deterministic forecasts a single value $x$ (in millimeters) for 24-hour rainfall at each location. This value is converted to a probability of $100\%$ if $x$ lies in the severity category of interest or a probability of $0\%$ otherwise. For example, the day+0 ECMWF deterministic forecast for Cairns was 57 mm, which lies below the MOD+ threshold of 297 mm, and so we have $\mathbb{P}(\text{MOD+}) = 0\%$.

The two ensemble prediction systems produce a single value $x$ (in millimeters) for 24-hour rainfall at each location for each ensemble member. The proportion of ensemble member values $x$ that lie in the severity category of interest could be used as the probability forecast. To illustrate, we consider the day+0 ECMWF ensemble forecast for Cairns. Rainfall predictions from the 51 ensemble members are shown as 51 black crosses on Fig. 8, while the AEP thresholds are indicated using grey dotted vertical lines. For the EXT category, 6 out of 51 ensemble members exceeded the EXT threshold of 415 mm, so $\mathbb{P}(\text{EXT}) = 6/51 \approx 12\%$, which lies in the 'possible' category.

Rather than calculating ensemble probabilities using a simple proportion of members in the severity category, instead we fit the empirical ensemble distribution to a parametric distribution, as shown with the orange curve in Fig. 8. The fitting method is described in Appendix A of Taggart et al. (2025), drawing on the work of Balakrishnan and Wang (2000). In the Cairns example, the probability of exceeding 415 mm based on the parametric fit is $15\%$, which also lies in the 'possible' category.

---

[3]In this section, forecasts and observations at Cairns are for Cairns Racecourse.

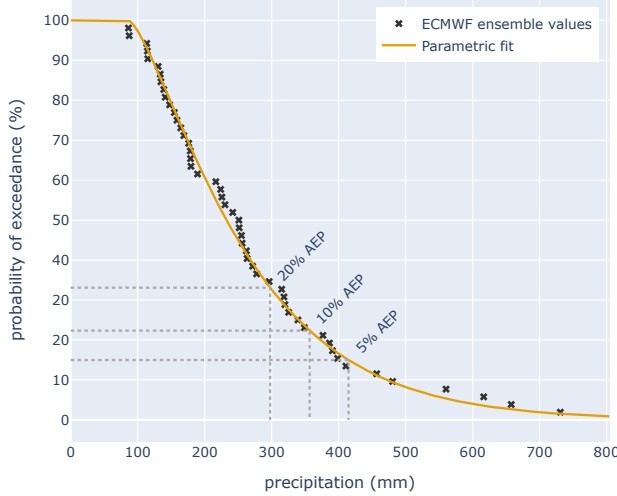

**Figure 8.** Probability of exceedance curve for daily precipitation on 17 December 2023 at Cairns, derived from the ECMWF ensemble forecast issued on early on the 17 December.

530 Fitting the data in this way gives the 18-member ACCESS-GE3 ensemble comparable resolution in probability space to the 51-member ECMWF ensemble, and avoids the over interpretation of forecast values from individual ensemble members, particularly for small ensembles.

 With that set-up, we now generate warning products for heavy rainfall for 17 December 2023. Figure 9 shows warning sequences generated by the ECWMF ensemble (top row), ECMWF deterministic (middle row) and ACCESS-GE3 ensemble

535 (bottom row), issued in the early hours of 15, 16 and 17 December (from left to right). The area of light blue shading indicates part of the Coral Sea whilst the grey lines show boundaries of districts that are used by the Bureau's forecast and warning service. The warnings issued on 15 December (left column) only contain a Yellow warning, because LONG-RANGE warning scaling is used. The other issue dates (centre and right columns) contain a mix of Yellow, Orange and Red Warnings. Note that the ECMWF deterministic forecast can only ever produce 'unlikely' or 'very likely' forecasts, and so at day+0 can never issue

540 a Yellow Warning due to the SHORT-RANGE warning scaling. The ECMWF Ensemble warning level at Cairns for day+0 is the Orange Warning, consistent with the PoE curve in Fig. 8.

## 4.2 Evaluation of the warning products

We now evaluate the predictive accuracy of the warning products. Precipitation accumulations for 17 December were observed over the domain of interest using a network of tipping bucket rain gauges. After automated and manual quality assurance

545 checks were conducted, 177 observations were obtained. Each observation was categorised as MIN, MOD, SEV and EXT according to warning service AEP thresholds (see Fig. 10). The observation at Cairns was 370 mm, which is categorised as SEV.

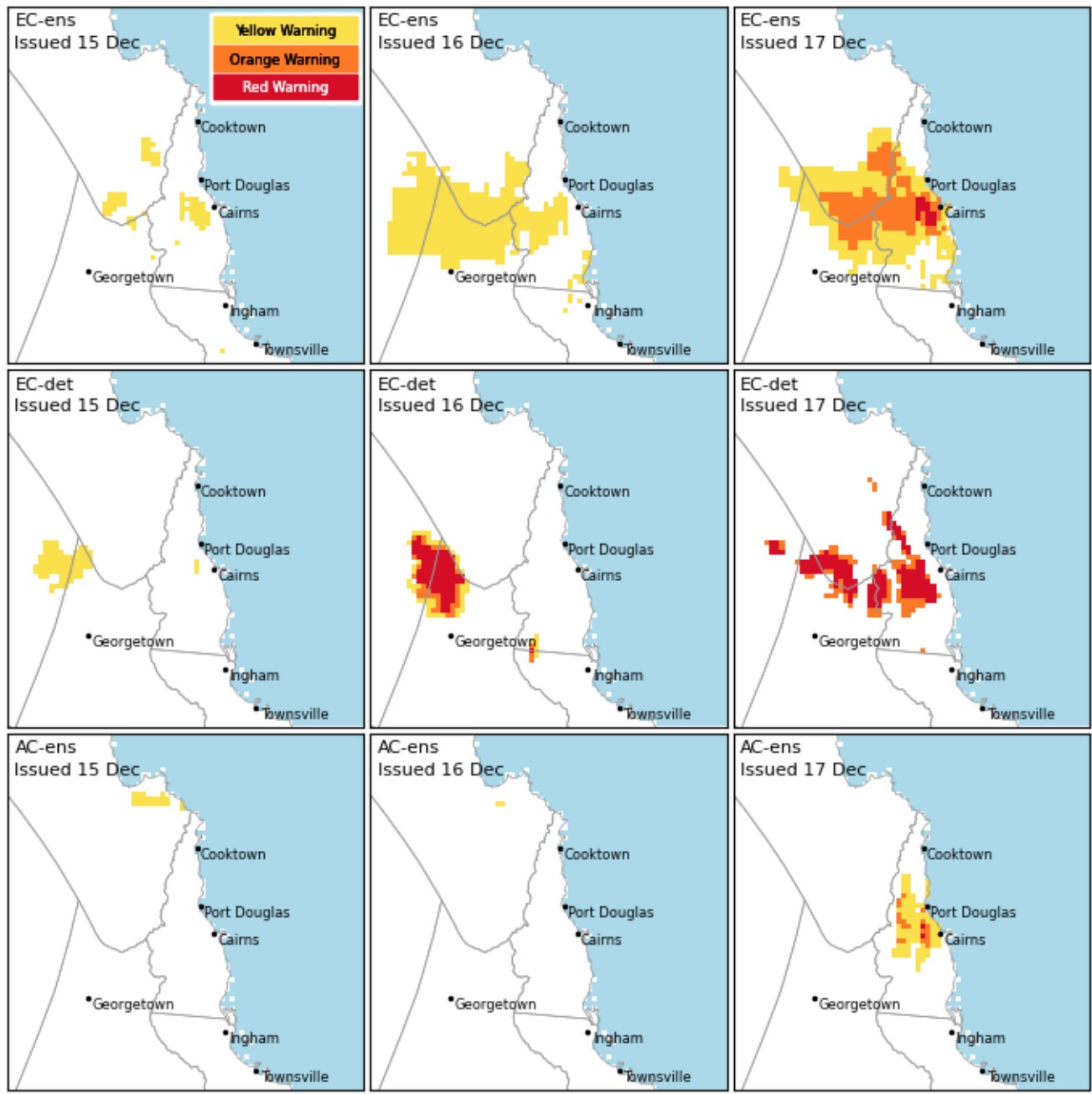

**Figure 9.** Warning sequences for 17 December 2023 as generated by the ECMWF ensemble (first row), ECMWF deterministic (second row) and ACCESS-GE3 ensemble (third row). Issuance times are the early morning of each issue date. From left to right, the warning scaling used follows the LONG-RANGE, MID-RANGE and SHORT-RANGE matrices shown in Fig. 4.

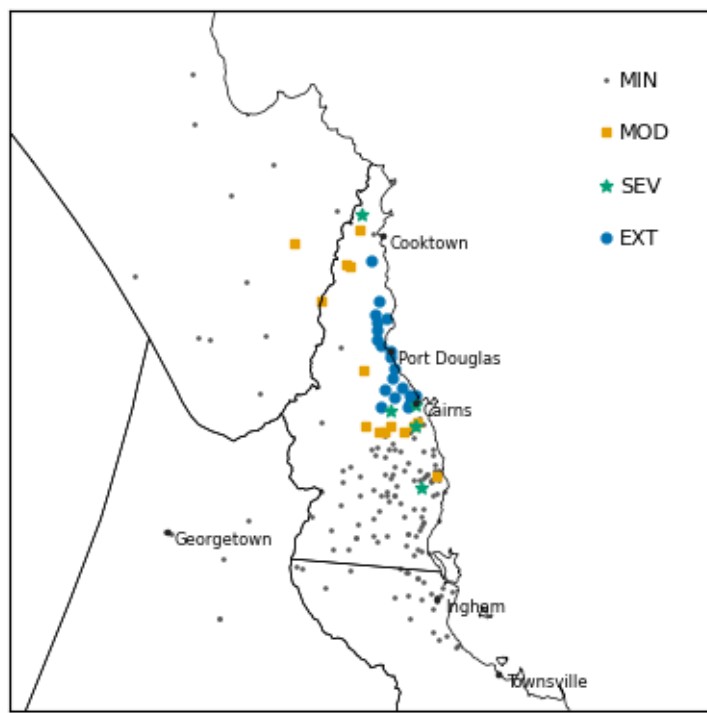

**Figure 10.** Observed 24-hour precipitation for 17 December 2024, categorised by severity.

We begin with a subjective comparison of the warning products with the observations and then apply an objective evaluation using the risk matrix score and warning score. Observations in the EXT category were concentrated along the coast and adjacent hinterland between Cairns and Cooktown. MOD and SEV observations were recorded a little further inland, and also a little to the south and north of the extreme observations. The remainder of the domain had observations in the MIN category. The warning products issued on 15 December generally did not produce a signal in the area of observed extreme rainfall, though there is signal inland of Cairns from the ECMWF ensemble. Both the ECMWF warning products issued on 16 December have a signal for heavy rain inland to the north of Georgetown. There are very few rain gauges in this area so whether this signal was accurate cannot be confirmed by data used in this study. Warnings issued early on 17 December all (correctly) have a signal inland of Cairns, while the warnings generated from the two ensembles also correctly include the coast between Cairns and Port Douglas. All three forecasts at all lead times mostly miss the heavy coastal rain observed between Port Douglas and Cooktown.

For an objective evaluation of the risk matrix forecasts we use the risk matrix score with equal emphasis placed on each decision point in the risk matrix (i.e., a uniform weight scheme with $w_{i,j} = 1$). For an objective evaluation of the warning products the warning score is used with with evaluation weights $(1, 1, 1)$ for their respective warning scalings. For each forecast system and issue time, forecasts are scored at each rain gauge and then the mean score across all rain gauges is calculated. For comparison, we also calculate the mean scores for the strategy 'Never warn.' By taking an unweighted mean across all

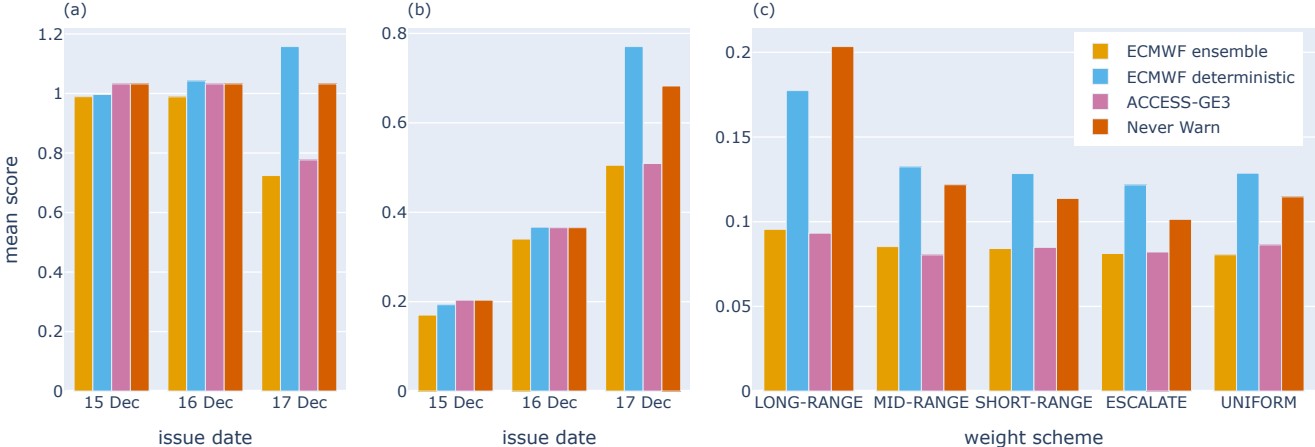

**Figure 11.** (a) Mean risk matrix scores $\overline{\text{RMaS}}$ for forecasts valid for 17 December 2023 by issue date, using a uniform weight scheme. (b) Mean warning scores $\overline{\text{WS}}$ for forecasts valid for 17 December 2023 by issue date. (c) Mean risk matrix scores $\overline{\text{RMaS}}$ for lead day+0 warnings valid for 17 December, each using a different weight scheme, with weights $w_{i,j}$ normalised to sum to 1. In all graphs, a lower mean score is better.

rain gauges, the mean score places a higher weight for predictive performance where the observation network is more dense.

One possible justification for this is that rain gauges are generally located where impacts (due to location of population, infrastructure or flood risk) are greater. If this emphasis is not desired, a weighted mean that places higher weight for more isolated observations could be used.

The mean risk matrix scores by issue date, using the uniform weight scheme, are shown in Fig. 11a. A lower mean score is better. For each issue date, the ECMWF ensemble performed best. The ECMWF ensemble forecast also improved as the

570 lead time decreased. In contrast, the ECMWF deterministic forecast did not improve with decreasing lead time, because the heaviest rainfall shifted eastward from an area without observations to an area with many observations in the MIN category. Both ensemble forecasts performed substantially better than the never warn strategy at lead day+0 (issue date of 17 December).

The mean warning scores by issue date are shown in Fig. 11b. Once again the ECMWF ensemble performed best on each issue date. The ACCESS-GE3 ensemble had the same score as the never warn strategy at days+2 and +1, which is unsurprising

given that its small warning footprint did not intersect with any rain gauges. ACCESS-GE3 outperformed the never warn strategy at day+0 by warning for areas where MOD to EXT observations were observed near Cairns and Port Douglas. The ECMWF deterministic warning product issued on 17 December performed poorly, giving a confident warning signal over many rain gauges to the southwest of Cairns where precipitation did not exceed 20% AEP. Note that the mean warning scores increase with decreasing lead time. This is because the warning scaling changes with lead time, and consequently the sum of

the weights $w_{i,j}$ used for scoring increases from 1 to 3 to 6 across lead days+2, +1 and +0.

Figure 11c shows mean risk matrix scores for day+0 forecasts issued early on 17 December based on different weight schemes. The LONG-RANGE, MID-RANGE and SHORT-RANGE weight schemes are the same as those in Table 4 with

$(v_1, v_2, v_3) = (1, 1, 1)$. The ESCALATE weight scheme uses the SHORT-RANGE scaling but with $(v_1, v_2, v_3) = (1, 2, 3)$. The UNIFORM weight scheme uses the weights $w_{i,j} = 1$. For each weight scheme, weights are normalised to sum to 1 to allow easier comparison between schemes. The results of Fig. 11c illustrate the general principle that forecaster rankings can depend on the choice of weight scheme (Ehm et al., 2016). For example, at day+0 ECMWF deterministic outperformed the 'Never warn' strategy when assessed using the LONG-RANGE weight scheme, but otherwise performed worse. Similarly, the ACCESS-GE3 ensemble had a lower mean score than the ECMWF ensemble using the LONG-RANGE scheme but not when using the UNIFORM scheme.

We reiterate the importance of selecting weights prior to any evaluation of forecasts and ideally so that forecasters know them in advance. The selected weights should reflect warning service priorities and should not be manipulated to obtain desirable rankings of forecasters.

Finally, we emphasise that the Tropical Cycle Jasper case study is merely designed to illustrate the warning framework and its consistent scoring methods. A superior public weather warning service for heavy rainfall would warn for intense rainfall over varying durations (say 30 minutes to 48 hours) and the period of intense rainfall could start any time of day (not just at the start of the local day). A warning would be issued if the period of intense rainfall (irrespective of duration) overlapped with the warning validity period. Moreover, evaluation of competing forecasters would need to be conducted over a long period, accompanied by tests for statistical significance, to conduct statistical inference regarding whether any forecast system was superior.

## 5   Summary and discussion

We have presented a general framework for determining warning levels for a hazard based on a probabilistic forecasts of the severity (or impact) of the hazard at a given warning lead time. The key tool used in the framework is the risk matrix, which gives a simple graphical representation of how a probabilistic forecast maps to a warning level, and is thus a useful device both for those who issue warning levels and those who wish to better understand or communicate them. While the use of risk matrices for the generation of warning levels is not new, a major contribution of this paper is to provide structure for their use that is free of issues found in other frameworks. These include perpetual warnings, undesirable warning level de-escalations and warning level determinations based on incomplete risk assessments. We emphasise that the framework is hazard agnostic, could be used for multi-hazard warnings (e.g., combined rainfall and wind), and could be used when severity categories are replaced with well-defined impact or consequence categories with the appropriate structural properties. Some simpler warning frameworks, such as those with a one-to-one correspondence between warning level and severity category and where warning issuance is defined by a single fixed probability threshold (Taggart et al., 2022), are special cases of the general framework.

Another major contribution of this paper is to furnish the warning framework with accompanying evaluation tools that are consistent with the aims of such warnings. The *risk matrix score* can be used to assess the predictive accuracy of risk matrix assessments. The *warning score*, which is a special case of the risk matrix score, assesses the predictive accuracy of warning levels by assessing the accuracy of risk matrix forecasts at those decision points that are critical to warning level determination.

Both scoring functions are *consistent* with the forecast directive to fill out the risk matrix assessment in-line with warning service specifications. This means that forecasters who make good probabilistic assessments, given the information at hand, and who comply with the forecast directive will be rewarded, while poorer probabilistic assessments or non-compliance is penalised. The availability of consistent scoring functions allows the possibility of a cycle of continuous improvement in warning accuracy through forecast system upgrades that are objectively measured to be better than their predecessors. Over the long term, these scores, or skill score versions of them (Potts, 2003, p. 27), can also be used as key performance indicator measures for warning accuracy.

To establish a warning service based on our framework, several considerations must be addressed. Firstly, severity and certainty categories should be defined to reflect the intention of the warning service and the capabilities of those responsible for its implementation. For example, severity categories could be directly linked to impact or consequence, while certainty categories could be chosen based on an assessment of the cost of misses relative to false alarms and consideration of forecast capabilities. Although the illustrative rainfall, flood and heat warning services in this article use four categories for both severity and certainty, any number of categories greater than one can be utilised.

Next, the warning scaling must be specified. Despite the growing shift towards impact-based warnings, there is limited research on the decision-making processes for determining the warning level (Jenkins et al., 2022). However, this is a crucial decision for the warning service which depends on several factors, especially the intended audience. Fundamentally, the scaling is informed by the required messaging. In Section 2.3, we introduced an extension to the framework that allows scaling to vary with lead time which may be desirable for warning systems that prioritise urgency or include messaging that depends on the time to hazard onset.

The above design considerations should be determined in collaboration with key stakeholders and users of the warning service (WMO, 2015; Jenkins et al., 2022). For example, understanding the relative implications of misses and false alarms for different user groups will inform the development of the warning system, which is likely to be an iterative process. Our verification approach allows for performance tracking, aiding in this endeavour. Similarly, public perception of 'flip-flops', wherein warnings are issued, cancelled and then re-instated for a location, should be considered. To maintain user trust in the warning service it may be beneficial to employ lower certainty thresholds for warning cancellation to minimise such sequences in situations where there is little variation in hazard probability. For public warnings of natural hazards, it is essential to adopt a people-centered approach which should leverage social and behavioural science research to ensure that warnings are communicated effectively, understood, and appropriately acted upon (WMO, 2022).

To improve message comprehension and public action, many international meteorological agencies have transitioned toward impact-based warnings (WMO, 2015). While this paper illustrates the framework using traditional meteorological hazards, the framework can be applied to impact warning services provided they are well-specified (e.g., an impact service considering quantitative damage forecasts for defined geographic areas). To verify such a system appropriately requires that suitable observations are available, which remains a challenge (Wyatt et al., 2023). Despite this, our approach provides a robust framework on which to build future quantitative impact warning services.

To implement the framework in practice, well-calibrated probabilistic forecast systems are highly desirable. However, the framework can still be applied in their absence by relying on deterministic models (as shown in our TC Jasper case study) or on more subjective likelihood assessments made by forecasters, if required. Nonetheless, strengthening probabilistic forecast system capabilities, such as through the development of statistically calibrated ensemble prediction systems, will greatly support warning decisions. Combined with observations, any real improvement in warning accuracy can now be objectively measured
to help support continued investment in such systems.

     For people to act on a warning they must receive it. Another benefit of our framework is that it simplifies the conversion of warning messages into the Common Alerting Protocol (CAP) format, which greatly enhances the reach of warnings. CAP is the international standard format for emergency alerting and public warning that allows consistent and comprehensible all-hazard emergency messages to be broadcast across a variety of communication systems (OASIS OPEN, 2024). To construct
a CAP message, critical information about the event is required, including the severity, certainty and urgency of the threat. The CAP severity and certainty can both be determined from our standard risk matrix provided it has been specified in a manner consistent with CAP definitions. CAP urgency, which represents the 'timeframe for responsive action', also requires information about the lead time to hazard onset, which could be accomodated by explicitly considering this factor in the framework, as described in Section 2.3.

Finally, the purpose of warning systems is to enable early action to save and protect lives, livelihoods and assets of people at risk (WMO, 2023). However, there is considerable global variation in the design and information conveyed by these systems, which can hinder community response and has led to recent calls for a unified approach (Neußner, 2021). The proposed framework, along with its evaluation method, provides a robust foundation for designing risk-based warning systems and stands as a strong candidate on which to base such an endeavour.

*Code and data availability.* A Python implementation of the risk matrix score, the algorithm for determining decision point weights for the warning score and an accompanying tutorial based on the synthetic experiment (https://scores.readthedocs.io/en/stable/tutorials/Risk_Matrix_Score.html), is available on the open source `scores` package (Leeuwenburg et al., 2024). A repository of data and code for the synthetic experiment and the Tropical Cyclone Jasper case study have been made available (Taggart, 2025).

## Appendix A: Consistency

We give a mathematical definition for a scoring function SF being consistent with the forecast directive, following the model of Gneiting (2011, Section 2.2), and then prove that the risk matrix score is consistent with the forecast directive. Here, scoring functions have negative orientation (i.e., a smaller score is better). The notation of Table 1 is assumed.

     Let $\mathcal{A}$ denote the $\sigma$-algebra on $\Omega$ generated by $\{\Omega_0, \ldots, \Omega_m\}$, noting that $\mathcal{A}$ is finite. Let $\mathcal{P}$ denote the space of probability measures on $\mathcal{A}$. A forecaster's probabilistic assessment for the warning service will be an element $P$ of $\mathcal{P}$, where $P(S_i)$ is the
680 probability that the outcome lies in severity category $S_i$.

Recall that $\mathcal{F}$ is the space of admissible forecasts for the nested severity categories. We define the operator $K : \mathcal{P} \to \mathcal{F}$ by $K(P) = (f_1, \ldots, f_m)$, where $f_i$ is the unique certainty category in $\mathcal{C}$ satisfying $P(S_i) \in f_i$. A forecaster with probabilistic assessment $P$ follows the forecast directive if they issue the categorical forecasts $K(P)$. Denote by $K_i(P)$ the $i$th component of $K(P)$.

**Definition.** We say that a scoring function $\mathrm{SF} : \mathcal{F} \times \Omega \to [0, \infty)$ is *consistent with the forecast directive* if

$$\mathbb{E}[\mathrm{SF}(K(P), Y)] \leq \mathbb{E}[\mathrm{SF}(G, Y)] \tag{A1}$$

for all probability distributions $P$ in $\mathcal{P}$ and all risk matrix forecasts $G$ in $\mathcal{F}$, where $Y$ is a random variable taking values in $\Omega$ with distribution $\mathbb{P}$ satisfying $\mathbb{P}(Y \in S_i) \in K_i(P)$ whenever $1 \leq i \leq m$.

**Lemma.** The elementary scoring function $\mathrm{ES}_{i,j}$ of Eq. (1) is consistent with the forecast directive.

*Proof.* Suppose that $P \in \mathcal{P}$, $G = (g_1, \ldots g_m) \in \mathcal{F}$ and that $Y$ is a random variable taking values in $\Omega$ with distribution $\mathbb{P}$ satisfying $\mathbb{P}(Y \in S_i) \in K_i(P)$ whenever $1 \leq i \leq m$. Let $f_i$ denote the certainty forecast $K_i(P)$ for the severity category $S_i$ and denote $P(S_i)$ by $p$, noting that $p \in f_i$. Then

$$\mathbb{E}[\mathrm{ES}_{i,j}(K(P), Y)] = \begin{cases} p_j P(\Omega \backslash S_i) & \text{if } f_i \text{ lies on or above } p_j, \\ (1 - p_j) P(S_i) & \text{if } f_i \text{ lies below } p_j. \end{cases}$$

$$= \begin{cases} p_j(1 - p) & \text{if } f_i \text{ lies on or above } p_j, \\ (1 - p_j) p & \text{if } f_i \text{ lies below } p_j. \end{cases}$$

Similarly,

$$\mathbb{E}[\mathrm{ES}_{i,j}(G, Y)] = \begin{cases} p_j(1 - p) & \text{if } g_i \text{ lies on or above } p_j, \\ (1 - p_j) p & \text{if } g_i \text{ lies below } p_j. \end{cases}$$

We aim to show Eq. (A1) for the scoring function $\mathrm{ES}_{i,j}$. We do so by considering two cases. In the case where $f_i$ lies on or above $p_j$, we have $p \geq p_j$. So if $g_i$ lies below $p_j$ then

$$\mathbb{E}[\mathrm{ES}_{i,j}(G, Y)] - \mathbb{E}[\mathrm{ES}_{i,j}(K(P), Y)] = p_j(1 - p) - (1 - p_j) p$$

$$= p_j - p$$

$$\geq 0.$$

On the other hand, if $g_i$ lies on or above $p_j$ then $\mathbb{E}[\mathrm{ES}_{i,j}(G, Y)] = \mathbb{E}[\mathrm{ES}_{i,j}(K(P), Y)]$. This establishes Eq. (A1). A similar argument establishes Eq. (A1) for the case where $f_i$ lies below $p_j$.

**Theorem.** The risk matrix scoring function RMaS of Eq. (2) is consistent with the forecast directive.

*Proof.* Suppose that $P \in \mathcal{P}$, $G = (g_1, \ldots g_m) \in \mathcal{F}$ and that $Y$ is a random variable taking values in $\Omega$ with distribution $\mathbb{P}$ satisfying $\mathbb{P}(Y \in S_i) \in K_i(P)$ whenever $1 \leq i \leq m$. Suppose that the weights $w_{i,j}$ are non-negative. Then

$$
\begin{aligned}
\mathbb{E}[\mathrm{RMaS}(K(P), Y)] &= \mathbb{E}\left[\sum_{i=1}^{m}\sum_{j=1}^{n} w_{i,j} \mathrm{ES}_{i,j}(K_i(P), Y)\right] \\
&= \sum_{i=1}^{m}\sum_{j=1}^{n} w_{i,j}\mathbb{E}[\mathrm{ES}_{i,j}(K(P), Y)] \\
&\leq \sum_{i=1}^{m}\sum_{j=1}^{n} w_{i,j}\mathbb{E}[\mathrm{ES}_{i,j}(G, Y)] \\
&= \mathbb{E}[\mathrm{RMaS}(G, Y)],
\end{aligned}
$$

where we have used linearity of the expectation operator and the lemma. This establishes Eq. (A1) for the scoring function RMaS.

## Appendix B: Algorithm for determining weighting

Given a warning service described by Table 1, the following algorithm gives values of the weights $w_{i,j}$ for the warning score
WS.

Algorithm

Set $w_{i,j} := 0$ whenever $1 \leq i \leq m$ and $1 \leq j \leq n$.

For warning level index $k$ in $(1, \ldots, q)$:

  Set $j_{\min} := n$.

For severity category index $i$ in $(1, \ldots, m)$:

    Set $J := \{r : \ell_k \preceq T(S_i, C_r), 1 \leq r \leq n\}$.

    If $J \neq \emptyset$ and $\min(J) < j_{\min}$:

      Set $j := \min(J)$, set $j_{\min} := j$ and set $w_{i,j} := w_{i,j} + v_k$.

  Return $(w_{i,j} : 1 \leq i \leq m, 1 \leq j \leq n)$.

End

  Essentially, $\min(J)$ is the probability threshold index at which a column in the scaling matrix switches from being below the warning level $\ell_k$ to being at or above the warning level $\ell_k$, if such a switch occurs. The variable $j_{\min}$ tracks whether such a switch occurred at the same probability threshold in a column to the left. If $\min(J) < j_{\min}$ it did not, and we can add a contribution of $v_k$ to the weight $w_{i,j}$. Switch points identify the warning level decision point within the column. The
730 justification for not adding a contribution of $v_k$ to the weight $w_{i,j}$ if this has already been done for a previous column at the same probability level $p_j$ is given by the following lemma.

**Lemma.** Fix the warning level index $k$ in $\{1, \ldots, q\}$ and the probability threshold index $j$ in $\{1, \ldots, n\}$ and suppose that there exist severity category indices $i'$ and $i$ satisfying $1 \leq i' < i \leq m$, $T(S_{i'}, C_{j-1}) \prec \ell_k \preceq T(S_{i'}, C_j)$ and $T(S_i, C_{j-1}) \prec$

$\ell_k \preceq T(S_i, C_j)$. If $(f_1, \ldots, f_m) \in \mathcal{F}$ and $\ell = \max\{T(S_{i'}, f_{i'}), T(S_i, f_i)\}$ then the column forecast $f_{i'}$ and warning scaling $T$ is sufficient to determine whether $\ell \prec \ell_k$ or $\ell_k \preceq \ell$.

*Proof.* Assume the hypotheses of the lemma and suppose that $(f_1, \ldots, f_m) \in \mathcal{F}$. Now if $\ell_k \preceq T(S_{i'}, f_{i'})$ then clearly $\ell_k \preceq \ell$.

On the other hand, if $T(S_{i'}, f_{i'}) \prec \ell_k$ then $f_{i'}$ must be a lower certainty category than $C_j$, by scaling property (b) and the hypothesis $T(S_{i'}, C_{j-1}) \prec \ell_k \preceq T(S_{i'}, C_j)$. Consequently, $f_i$ also must be a lower certainty category than $C_j$, since $f_i$ cannot be higher than $f_{i'}$ by the fact that $(f_1, \ldots, f_m) \in \mathcal{F}$. So $T(S_i, f_i) \preceq T(S_i, C_{j-1}) \prec \ell_k$, whence $\ell \prec \ell_k$.

*Author contributions.* Both authors contributed to the conceptualisation of ideas for this paper and to its writing, both in draft and final stages. Data curation, software development, data analysis and creation of figures were conducted by RJT.

*Competing interests.* The authors declare that they have no conflict of interest.

*Acknowledgements.* The authors would like to acknowledge Beth Ebert, Deryn Griffiths and Samar Momin for feedback on an earlier version of this manuscript. Nicholas Loveday, Tennessee Leeuwenburg and Stephanie Chong gave code reviews of the Python implementation of the risk matrix score in the open-source `scores` package. Thanks to Carla Mooney for her valuable guidance on the social science aspects of this work, and to Rob Johnson for making forecast data for TC Jasper available.

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
