# Peer review of "Warnings based on risk matrices: a coherent framework with consistent evaluation"

_EGUsphere, 2025_

## Author Response (AR1)

**Authors' response**

Only the first reviewer made suggestions for improving the manuscript. Relevant comments from this reviewer are reproduced below in bold font, with our response in non-bold font.

We have also taken the liberty of making a few minor changes to the document to fix a handful of typos and update the acknowledgements.

**1. Terminology and Framing:**

**While the mathematical rigor is a strength, early sections could benefit from briefly reinforcing why these inconsistencies in risk matrices matter for public safety and policy credibility. Consider simplifying the initial explanation of "forecast directive" and "warning directive" for non-technical readers.**

We think the best way of improving the initial explanation of directives is to provide a simple illustrative example. We have done this in Lines 26-27 of the revised text:

> An example of a forecast directive for a warning service for damaging wind gusts is "Issue a warning if and only if the probability of a wind gust exceeding 90 km/h is at least 10%".

**2. Comparison with Existing Systems:**

**The distinction from the UK Met Office (UKMO) and other operational frameworks is clear, but it might help to include a side-by-side visual comparison in an appendix or supplementary material (if possible).**

We agree that a side-by-side visual comparison will be helpful for the reader. It fits naturally in Section 2.2. See Figure 3 (which is new) and some additional text which refers to the figure (Lines 160-161, 184).

**3. Evaluation Weights:**

**The method for deriving weights from stakeholder input (e.g., community consultation on false alarm vs. miss costs) is strong. However, a brief reflection on the subjectivity and variability in such consultations would add depth.**

We believe that such discussion on this is beyond the scope of the current work. However, we have included the following sentence at the end of Section 3.1 (Lines 312-314):

> Although the process for determining weights in this fictitious flood example was presented straightforwardly, this framework motivates further research into developing best practices for eliciting thresholds and weights through stakeholder consultation.

**4. Scalability to Multi-Hazard Systems:**

**Although the framework is hazard-agnostic, a discussion on how it could scale or adapt to multi-hazard interactions (e.g., flood + wind) would strengthen its applicability. That being said, it would be helpful to shed light on this framework toward earthquake hazards as they are growing in frequency (if possible).**

For multi-hazard interactions, we have added the additional discussion (Lines 150-152):

> More generally, the framework could be applied to an index, which itself represents complex multi-hazard interactions. An example of such an index is the Fire Behaviour Index (FBI) used in the Australian Fire Danger Ratings System (AFDRS), which combines weather and fuel state information to determine the severity of fire behaviour (Hollis et al., 2024).

This is in addition to the simple multi-hazard example provided in the original submission for hailstones and wind gusts (Lines 149-149).

Although the framework is applicable to earthquake hazards, we believe it is not appropriate to discuss this in detail, as earthquakes lie outside the authors' area of expertise.

**5. Lead Time Scaling:**

**The use of distinct matrices for LONG-, MID-, and SHORT-range phases is excellent. It would be helpful to mention how this could be dynamically updated as new ensemble data arrives.**

How the arrival of new ensemble data impacts the warning issue process will depend on the way each warning service is designed. Going into such details is beyond the scope of this manuscript but could be explored using concrete warning service examples in a follow-up paper. In our response to the reviewer, we gave further thoughts on how this might play out. We reproduce the response below, noting that we have not changed the manuscript in responding to this particular comment.

> Nonetheless, we note here that there are at least two factors at play. One is where the lead time phases are a function of the onset to severe phenomena, and new ensemble data shifts the time of onset sufficiently to change the phase. The other is where new ensemble data leads to a re-evaluation of the likelihood and/or severity of the phenomena, which may prompt an update of the warning based on pre-defined amendment criteria for the warning service.